# Magnifiers in Some Generalization of the Full Transformation Semigroups

**Thananya Kaewnoi [1], Montakarn Petapirak [2] and Ronnason Chinram [2,3,\*]**

[1]  Department of Mathematics and Statistics, Prince of Songkla University, Hat Yai, Songkhla 90110, Thailand; thananya.k2538@gmail.com

[2]  Algebra and Applications Research Unit, Department of Mathematics and Statistics, Prince of Songkla University, Hat Yai, Songkhla 90110, Thailand; montakarn.p@psu.ac.th

[3]  Centre of Excellence in Mathematics, CHE, Si Ayuthaya Road, Bangkok 10400, Thailand

\*  Correspondence: ronnason.c@psu.ac.th

**Abstract:** An element $a$ of a semigroup $S$ is called a left [right] magnifier if there exists a proper subset $M$ of $S$ such that $aM = S$ ($Ma = S$). Let $T(X)$ denote the semigroup of all transformations on a nonempty set $X$ under the composition of functions, $\mathcal{P} = \{X_i \mid i \in \Lambda\}$ be a partition, and $\rho$ be an equivalence relation on the set $X$. In this paper, we focus on the properties of magnifiers of the set $T_\rho(X, \mathcal{P}) = \{f \in T(X) \mid \forall (x, y) \in \rho, (xf, yf) \in \rho \text{ and } X_i f \subseteq X_i \text{ for all } i \in \Lambda\}$, which is a subsemigroup of $T(X)$, and provide the necessary and sufficient conditions for elements in $T_\rho(X, \mathcal{P})$ to be left or right magnifiers.

**Keywords:** magnifiers; magnifying elements; transformation semigroups; equivalence relations; partitions

## 1. Introduction

Let $T(X)$ denote the semigroup of all transformations on a nonempty set $X$. We will study magnifiers or magnifying elements in the generalization of this semigroup. Firstly, we give some concepts of the magnifiers in a semigroup. This notion was originally proposed by Ljapin in [1]. An element $a$ of a semigroup $S$ is called a left (right) magnifier if $aM = S$ ($Ma = S$) for some proper subset $M$ of $S$. In 1992, the existence of magnifiers in simple and bisimple semigroups were established by Catino and Migliorini [2]. After that, K.D. Magill, Jr. [3] introduced nonstrong and strong magnifiers, but he focused on a nonstrong magnifier which is simply called a magnifier. The conditions for elements in any subsemigroup of $T(X)$ with the identity to be a magnifier were established in his paper; moreover, he applied the result in some specific transformation semigroups. In 1996, Gutan [4] studied strong and nonstrong magnifiers in a semigroup and constructed the semigroup containing both left strong and left nonstrong magnifiers which answers the question that was raised by K.D. Magill, Jr. in [3]. A year later, he proved in [5] that every semigroup containing magnifiers is factorizable. Recently, the study of magnifiers in semigroups have been carried out by many authors. For instance, Chinram, Petchkaew, and Buapradist investigated the magnifiers of $T(X)$ determined by a partition of a set $X$ in [6]. Left and right magnifiers in the partial transformation semigroups were characterized by Luangchaisri, Changphas, and Phanlert in [7].

Thus far, many authors have extensively studied the transformation semigroups that preserve an equivalence relation. In 2004, Araujo and Konieczny [8] investigated a subsemigroup $T(X, \rho, R)$ of $T(X)$, where

$$T(X, \rho, R) = \{a \in T(X) \mid Ra \subseteq R \text{ and } (x, y) \in \rho \text{ implies } (xa, ya) \in \rho\}$$

consisting of all transformations preserving both an equivalence relation $\rho$ on $X$ and a cross-section $R$ of the partition $X/\rho$ determined by $\rho$. Moreover, they characterized the equivalence relation $\rho$ on $X$ for this semigroup to be regular and described the equivalence relation $\rho$ on $X$ leading to $\mathcal{D} = \mathcal{J}$ in the semigroup $T(X, \rho, R)$, where $\mathcal{D}$ and $\mathcal{J}$ are Green's relations. Later, Pei [9] examined the subsemigroup $T_\rho(X)$ of $T(X)$, which is defined as

$$T_\rho(X) = \{f \in T(X) \mid \forall(x, y) \in \rho, (f(x), f(y)) \in \rho\}$$

where $\rho$ is an equivalence relation. This author investigated the necessary and sufficient conditions of $T_\rho(X)$ to be a regular semigroup and provided the conditions for any two elements in $T_\rho(X)$ having finite ranges to be $\mathcal{J}$-equivalent where $\mathcal{J}$ is a Green's relation. Meanwhile, Pei and Zou characterized Green's equivalences in some subsemigroups of $T_\rho(X)$ in [10]. In 2009, Pei and Deng [11] exposed the conditions for the equivalence relation $\rho$ on an infinite set $X$, which leads to $\mathcal{D} = \mathcal{J}$ in the semigroup $T_\rho(X)$. In [12], Sun and Pei gave necessary and sufficient conditions of $T_{\rho_1}(X) \cap T_{\rho_2}(X)$ for equivalence relations $\rho_1$ and $\rho_2$ on X such that $\rho_2 \subseteq \rho_1$ under the assumption that $\rho_1$ and $\rho_2$ are comparable to be regular. In addition, naturally ordered semigroups of full and partial transformations preserving an equivalence relation were studied in [13] and [14]. Previously, the authors published the conditions of elements being left and right magnifiers in the full and partial transformation semigroups which preserve an equivalence relation in [15] and [16], respectively. As we can see, Sun and Pei [12] studied Green's relations on $T(X)$ preserving two equivalence relations. However, no one has yet studied the magnifiers in this semigroup. Thus, adding the condition preserving the partition on a set $X$ would be a useful first step for the future work that is involved.

From now on, let $\rho$ be an equivalence relation on a nonempty set $X$ and denote a partition on a nonempty set $X$ by $\mathcal{P} = \{X_i \mid i \in \Lambda\}$, a collection of nonempty subsets of $X$ satisfying $X = \bigcup_{i \in \Lambda} X_i$ with $X_i \cap X_j = \varnothing$ for all $i, j \in \Lambda$ such that $i \neq j$. For any $f, g \in T_\rho(X)$ and $x \in X$, the notations $xf$ and $xfg$ are used instead of $f(x)$ and $(g \circ f)(x)$, respectively. The image of $f$ is denoted by ran $f$. Then, we define

$$T_\rho(X, \mathcal{P}) = \{f \in T(X) \mid \forall(x, y) \in \rho, (xf, yf) \in \rho \text{ and } X_i f \subseteq X_i \text{ for all } i \in \Lambda\},$$

for any $x \in X$, we denote the equivalence class of $\rho$ containing $x$ by $[x]_\rho = \{y \in X \mid x\rho y\}$. Let $X/\rho = \{[x]_\rho \mid x \in X\}$ and $(X_i, x_j) = X_i \cap [x_j]_\rho$ for $x_j \in X$ and $X_i \in \mathcal{P}$. The main purpose of this paper is to go a further step by considering magnifiers in $T_\rho(X, \mathcal{P})$ which preserves both an equivalence relation $\rho$ and a partition $\mathcal{P}$ on X. In particular, it is verified that a magnifier exists if and only if at least one element of $\mathcal{P}$ is finite. Futhermore, the necessary and sufficient conditions for elements in $T_\rho(X, \mathcal{P})$ to be magnifiers are established.

## 2. Left Magnifiers in $T_\rho(X, P)$

In this section, we begin with some properties of left magnifiers in $T_\rho(X, P)$ and the conditions for elements in this semigroup to be a left magnifier are provided.

**Lemma 1.** *If $f$ is a left magnifier in $T_\rho(X, \mathcal{P})$, then $f$ is injective.*

**Proof.** By assumption, let $M \subsetneq T_\rho(X, \mathcal{P})$ be such that $fM = T_\rho(X, \mathcal{P})$. Clearly, the identity map $id_X$ on X belongs to $T_\rho(X, \mathcal{P})$. Thus, there exists a function $h \in M$ such that $fh = id_X$. This implies that $f$ is injective. $\square$

However, the converse of Lemma 1 is not true since there is no proper subset $M$ of $T_\rho(X, \mathcal{P})$ such that $id_X M = T_\rho(X, \mathcal{P})$.

**Lemma 2.** *Let $f$ be a left magnifier in $T_\rho(X, \mathcal{P})$. For any $x, y \in X$, if $(xf, yf) \in \rho$, then $(x, y) \in \rho$.*

**Proof.** Let $f$ be a left magnifier in $T_\rho(X, \mathcal{P})$ and $M \subsetneq T_\rho(X, \mathcal{P})$ be such that $fM = T_\rho(X, \mathcal{P})$. Then, $fh = id_X$ for some $h \in M$. Let $x, y \in X$ be such that $(xf, yf) \in \rho$. Hence, $(x, y) = (xfh, yfh) = ((xf)h, (yf)h)$. Therefore, $(x, y) \in \rho$ since $h \in T_\rho(X, \mathcal{P})$ and $(xf, yf) \in \rho$. $\square$

Note that, if $f \in T_\rho(X, \mathcal{P})$, then, for any $x \in X$ and $X_i \in \mathcal{P}$, $xf \in X_i$ implies $x \in X_i$.

**Lemma 3.** *If $f \in T_\rho(X, \mathcal{P})$ is bijective, then $f$ is not a left magnifier in $T_\rho(X, \mathcal{P})$.*

**Proof.** Let $f \in T_\rho(X, \mathcal{P})$ be bijective. Then, $f^{-1}$ is also bijective. Suppose that $f$ is a left magnifier. Then, there is a proper subset $M$ of $T_\rho(X, \mathcal{P})$ such that $fM = T_\rho(X, \mathcal{P})$. Clearly, $fM = fT_\rho(X, \mathcal{P})$. To show that $f^{-1} \in T_\rho(X, \mathcal{P})$, let $x, y \in X$ be such that $(x, y) \in \rho$. Since $f$ is surjective and by Lemma 2, there exists $(a, b) \in \rho$ such that $af = x$ and $bf = y$. Then, $(xf^{-1}, yf^{-1}) = (a, b) \in \rho$. Next, let $x \in X_i$ for some $X_i \in \mathcal{P}$. Since $f$ is surjective, there exists $a \in X_i$ such that $af = x$. Then, $xf^{-1} = a \in X_i$. Therefore, $f^{-1} \in T_\rho(X, \mathcal{P})$. Hence, $M = f^{-1}fM = f^{-1}fT_\rho(X, \mathcal{P}) = T_\rho(X, \mathcal{P})$, which is a contradiction. $\square$

By Lemmas 1 and 3, we obtain the following corollary.

**Corollary 1.** *If $f$ is a left magnifier in $T_\rho(X, \mathcal{P})$, then $f$ is injective but not surjective.*

**Lemma 4.** *Let $\mathcal{P} = \{X_i \mid i \in \Lambda\}$ be a partition on a set $X$. If $X_i \in \mathcal{P}$ is finite for all $i \in \Lambda$, then there exists no left magnifier in $T_\rho(X, \mathcal{P})$.*

**Proof.** Suppose that there is a left magnifier $f$ in $T_\rho(X, \mathcal{P})$. By Lemma 1, $f|_{X_i}$ is bijective for all $i \in \Lambda$. Since $Xf = \left( \bigcup_{i \in \Lambda} X_i \right) f = \bigcup_{i \in \Lambda} X_i f = \bigcup_{i \in \Lambda} X_i = X$, $f$ is surjective which is a contradiction. $\square$

It is noticeable in Lemma 4 that if a left magnifier exists in $T_\rho(X, \mathcal{P})$, then $X_i$ is infinite for some $i \in \Lambda$. However, the converse of this statement is not true in general. It is illustrated by the following counterexample.

**Example 1.** *Let $X = \mathbb{Z}$ and $\mathcal{P} = \{X_i \mid i \in \mathbb{N} \cup \{0\}\}$ where $X_0 = \{0, -1, -2, \ldots\}$ and $X_i = \{2i - 1, 2i\}$ for all $i \in \mathbb{N}$. Define a relation $\rho$ on $X$ by $\rho = \bigcup_{j=1}^{\infty} (A_j \times A_j)$ where $A_1 = \{0, \pm 1, \pm 2\}$ and $A_j = \{\pm 2j, \pm(2j - 1)\}$ for all $j \in \mathbb{N}$ and $j \geq 2$. Clearly, $(X_i, x)$ is finite for all $i \in \mathbb{N} \cup \{0\}$ and $x \in X$. Note that $(X_0, x_i) \subseteq A_i$ and $X_i \subseteq A_i$; moreover, $A_i = (X_0, x_i) \sqcup X_i$ where $[x_i]_\rho = A_i$ for all $i \in \mathbb{N}$. Let $f$ be an injection in $T_\rho(X, \mathcal{P})$. Then, $X_i f = X_i \subseteq A_i$ for all $i \in \mathbb{N}$. This forces that $(X_0, x_i)f = (X_0, x_i) \subseteq A_i$ where $[x_i]_\rho = A_i$ for all $i \in \mathbb{N}$. Then, $f$ is a surjection. This implies that there exists no left magnifier in $T_\rho(X, \mathcal{P})$.*

**Corollary 2.** *If $X$ is a finite set, then $T_\rho(X, \mathcal{P})$ has no left magnifiers.*

We are now going to work under the assumption that, for each $x \in X$, there is exactly one $X_i \in \mathcal{P}$ such that $[x]_\rho \subseteq X_i$. That is, the partition associated with the equivalence relation $\rho$ is a refinement of $\mathcal{P}$.

**Lemma 5.** *Let $\mathcal{P} = \{X_i \mid i \in \Lambda\}$ be a partition on a set $X$ such that $X_i$ is infinite for some $i \in \Lambda$ and $X/\rho$ is a refinement of $\mathcal{P}$. If $f \in T_\rho(X, \mathcal{P})$ is injective but not surjective and for any $x, y \in X$, $(xf, yf) \in \rho$ implies $(x, y) \in \rho$, then $f$ is a left magnifier of $T_\rho(X, \mathcal{P})$.*

**Proof.** Assume that $f \in T_\rho(X, \mathcal{P})$ is injective but not surjective. For each $x \in \operatorname{ran} f$, there is $y_x \in X$ such that $y_x f = x$.
  **Case 1.** $|[x]_\rho| = 1$ for all $x \in X$.

Let $M = \{h \in T_\rho(X, \mathcal{P}) \mid xh = x$ for all $x \notin \operatorname{ran} f\}$. Clearly, $M$ is a proper subset of $T_\rho(X, \mathcal{P})$. For any function $g$ in $T_\rho(X, \mathcal{P})$, define a function $h \in T_\rho(X, \mathcal{P})$ by

$$
xh = \begin{cases} y_x g & \text{if } x \in \operatorname{ran} f, \\ x & \text{if } x \notin \operatorname{ran} f \end{cases}
$$

for all $x \in X$. Clearly, $h \in M$. For any $x \in X$, $xfh = y_{xf}g$. Since $y_{xf}f = xf$ and $f$ is injective, $y_{xf} = x$. Therefore, $xfh = xg$. This shows that $fh = g$, which implies that $fM = T_\rho(X, \mathcal{P})$.

**Case 2.** $|[x]_\rho| > 1$ for some $x \in X$.

Let $X = \{x_j \mid j \in \Lambda'\}$. For any $i \in \Lambda$ and $j \in \Lambda'$ with $(X_i, x_j) \cap \operatorname{ran} f \neq \varnothing$, choose $x_{ij} \in (X_i, x_j) \cap \operatorname{ran} f$. Then, there exist $y_{x_{ij}} \in X_i$ such that $y_{x_{ij}}f = x_{ij}$. If $(X_i, x_j) \cap \operatorname{ran} f = \varnothing$, choose $x'_{ij} \in (X_i, x_j)$. Let $M = \{h \in T_\rho(X, \mathcal{P}) \mid h$ is not injective$\}$. Clearly, $M$ is a proper subset of $T_\rho(X, \mathcal{P})$. For any function $g$ in $T_\rho(X, \mathcal{P})$, define a function $h \in T_\rho(X, \mathcal{P})$ for all $x \in X$ by

$$
xh = \begin{cases} y_x g & \text{if } x \in \operatorname{ran} f, \\ y_{x_{ij}}g & \text{if } x \notin \operatorname{ran} f \text{ and } x \in (X_i, x_j) \text{ such that } (X_i, x_j) \cap \operatorname{ran} f \neq \varnothing, \\ x'_{ij} & \text{if } x \notin \operatorname{ran} f \text{ and } x \in (X_i, x_j) \text{ such that } (X_i, x_j) \cap \operatorname{ran} f = \varnothing. \end{cases}
$$

Let $(a, b) \in \rho$. Then, $a, b \in [x_j]_\rho$ for some $x_j \in X$. If $a, b \in \operatorname{ran} f$, then there exist $y_a, y_b \in X$ such that $y_a f = a$ and $y_b f = b$. By assumption, $(y_a, y_b) \in \rho$. Hence, $(ah, bh) = (y_a g, y_b g) \in \rho$. Since $X/\rho$ is a refinement of $\mathcal{P}$, $a$ and $b$ must belong to $X_i$ for some $i \in \Lambda$. Thus, $a, b \in (X_i, x_j)$. If $a, b \notin \operatorname{ran} f$, then we have two cases to consider. It is clear that $(ah, bh) = (y_{x_{ij}}g, y_{x_{ij}}g) \in \rho$ if $(X_i, x_j) \cap \operatorname{ran} f \neq \varnothing$ and $(ah, bh) = (x'_{ij}, x'_{ij}) \in \rho$ if $(X_i, x_j) \cap \operatorname{ran} f = \varnothing$. Next, we may assume that $a \in \operatorname{ran} f$ and $b \notin \operatorname{ran} f$. Then, we choose $x_{ij} = a \in (X_i, x_j) \cap \operatorname{ran} f$ and hence $(ah, bh) = (y_a g, y_a g) \in \rho$. Hence, it is readily seen that $h$ preserves the equivalence relation $\rho$. Moreover, it is easy to see that $h$ preserves the partition $\mathcal{P}$ on $X$ as well. Since $[x]_\rho > 1$ for some $x \in X$ and $f$ is injective but not surjective, there exists $x_0 \notin \operatorname{ran} f$. Then, $x_0 f \in (X \setminus \{x_0\})h$ and hence $h$ is not injective. Therefore, $h \in M$. For any $x \in X$, $xfh = y_{xf}g$. Again, we obtain $y_{xf} = x$. Therefore, $xfh = xg$. This shows that $fh = g$, which implies that $fM = T_\rho(X, \mathcal{P})$. $\square$

The following examples illuminate the ideas of the proof given in Lemma 5.

**Example 2.** *Let $X = \mathbb{Q}$ and $\mathcal{P}$ be a partition on $X$ such that $\mathcal{P} = \{\mathbb{Q}^+, \mathbb{Q}^- \cup \{0\}\}$. For each $n \in \mathbb{N}$, denote $A_n = \left\{\dfrac{x}{n} \mid x \in \mathbb{N} \text{ and } \gcd(x, n) = 1\right\}$ and $\mathcal{A} = \mathbb{Q}^- \setminus \{-2n + 1 \mid n \in \mathbb{N}\}$. Define a relation $\rho$ on $X$ by $\rho = \bigcup\limits_{i=1}^{\infty}(A_{2i} \times A_{2i}) \cup \bigcup\limits_{i=1}^{\infty}\left(A_{2i-1} \cup \{-2i + 1\} \times A_{2i-1} \cup \{-2i + 1\}\right) \cup (\mathcal{B} \times \mathcal{B})$ where $\mathcal{B} = \mathcal{A} \cup \{0\}$. Thus, $\rho$ is an equivalence relation on $X$ and $\mathbb{Q}^+ \in \mathcal{P}$ is infinite. Clearly, $\mathcal{B}$ and $A_n$ are infinite for all $n \in \mathbb{N}$. Then, there exist the bijective functions $\varphi_1 : \mathcal{B} \longrightarrow \mathcal{B}$, $\varphi_2 : A_2 \longrightarrow A_1$, $\varphi_{2n} : A_{2n} \longrightarrow A_{2(n-1)}$ for all $n \geq 2$ and $\varphi_{2n-1} : A_{2n-1} \longrightarrow A_{2(n+1)-1}$ for all $n \in \mathbb{N}$. Define a function $f \in T_\rho(X, \mathcal{P})$ by*

$$
xf = \begin{cases} x\varphi_i & \text{if } x \in A_i, i \in \mathbb{N}, \\ x - 2 & \text{if } x \in \{-2n + 1 \mid n \in \mathbb{N}\}, \\ x & \text{otherwise.} \end{cases}
$$

*Clearly, $-1 \notin \operatorname{ran} f$. Therefore, $f$ is injective but not surjective and, for any $x, y \in X$, $(xf, yf) \in \rho$ implies $(x, y) \in \rho$ but $f$ is not a left magnifier of $T_\rho(X, \mathcal{P})$ since there is no function $h \in T_\rho(X, \mathcal{P})$ such that $fh = id_X$.*

**Example 3.** *Let $X = \mathbb{Z}$ and $\mathcal{P}$ be a partition on $X$ such that $\mathcal{P} = \{X_1, X_2\}$ where $X_1 = \{0, -1, -2, -3, \dots\}$ and $X_2 = \{1, 2, 3, \dots\}$. Define a relation $\rho$ on $X$ by $\rho = \bigcup_{i=0}^{\infty} (A_i \times A_i)$ where $A_0 = X_1, A_1 = \{1\}, A_2 = \{2, 3\}, A_3 = \{4, 5, 6\}, A_4 = \{7, 8, 9, 10\}, \dots$ It is obvious that $\rho$ is an equivalence relation on $X$. Clearly, $X/\rho = \{\{0, -1, -2, -3, \dots\}, \{1\}, \{2, 3\}, \{4, 5, 6\}, \{7, 8, 9, 10\}, \dots\}$ and $X_1 \in \mathcal{P}$ is infinite. Define a function $f$ on $X$ by*

$$xf = \begin{cases} x & \text{if } x \in A_0, \\ x + i & \text{if } x \in A_i, i > 0. \end{cases}$$

*For convenience, we write $f$ as*

$$f = \begin{pmatrix} \cdots & -3 & -2 & -1 & 0 & 1 & 2 & 3 & 4 & 5 & 6 & 7 & 8 & 9 & 10 & \cdots \\ \cdots & -3 & -2 & -1 & 0 & 2 & 4 & 5 & 7 & 8 & 9 & 11 & 12 & 13 & 14 & \cdots \end{pmatrix}.$$

*It is noticeable that $f$ is injective but not surjective and for any $x, y \in X$, $(xf, yf) \in \rho$ implies $(x, y) \in \rho$. Let $M = \{h \in T_\rho(X, \mathcal{P}) \mid h \text{ is not injective}\}$ and consider the element $g$ of $T_\rho(X, \mathcal{P})$, which is defined by*

$$xg = \begin{cases} x & \text{if } x \in A_i, i \leq 2, \\ x + i & \text{if } x \in A_i, i > 2. \end{cases}$$

*Thus, we write $g$ as*

$$g = \begin{pmatrix} \cdots & -3 & -2 & -1 & 0 & 1 & 2 & 3 & 4 & 5 & 6 & 7 & 8 & 9 & 10 & \cdots \\ \cdots & -3 & -2 & -1 & 0 & 1 & 2 & 3 & 7 & 8 & 9 & 11 & 12 & 13 & 14 & \cdots \end{pmatrix}.$$

*By Lemma 5, $f$ is a left magnifier and there exists an element $h \in M$ such that $fh = g$. Note that, for all $j \geq 1$, $A_j \cap \text{ran } f \neq A_j$ and hence there is an element $x_j \in A_j$ and $x_j \notin \text{ran } f$. To get the desired result, for any $x_j \notin \text{ran } f$ such that $(X_i, x_j) \cap \text{ran } f \neq \varnothing$, choose $y_{x_j} = \min\left((X_i, x_j) \cap \text{ran } f\right) = \min A_j$ for all $j \geq 4$ and define a function $h$ in $T_\rho(X, \mathcal{P})$ by $xh = x$ for all $x \in A_0 \cup A_1$, $2h = 3h = 1$, $4h = 6h = 2$, $5h = 3$ and*

$$xh = \begin{cases} x & \text{if } x \neq \max A_j, \\ y_{x_j} & \text{if } x = \max A_j \end{cases}$$

*for all $x \in A_j$ such that $j \geq 4$. For convenience, we write $h$ as*

$$h = \begin{pmatrix} \cdots & -3 & -2 & -1 & 0 & 1 & 2 & 3 & 4 & 5 & 6 & 7 & 8 & 9 & 10 & \cdots \\ \cdots & -3 & -2 & -1 & 0 & 1 & 1 & 1 & 2 & 3 & 2 & 7 & 8 & 9 & 7 & \cdots \end{pmatrix}.$$

*Clearly, $h \in M$. Therefore, we have*

$$fh = \begin{pmatrix} \cdots & -3 & -2 & -1 & 0 & 1 & 2 & 3 & 4 & 5 & 6 & 7 & 8 & 9 & 10 & \cdots \\ \cdots & -3 & -2 & -1 & 0 & 2 & 4 & 5 & 7 & 8 & 9 & 11 & 12 & 13 & 14 & \cdots \end{pmatrix}$$

$$\begin{pmatrix} \cdots & -3 & -2 & -1 & 0 & 1 & 2 & 3 & 4 & 5 & 6 & 7 & 8 & 9 & 10 & \cdots \\ \cdots & -3 & -2 & -1 & 0 & 1 & 1 & 1 & 2 & 3 & 2 & 7 & 8 & 9 & 7 & \cdots \end{pmatrix}$$

$$= \begin{pmatrix} \cdots & -3 & -2 & -1 & 0 & 1 & 2 & 3 & 4 & 5 & 6 & 7 & 8 & 9 & 10 & \cdots \\ \cdots & -3 & -2 & -1 & 0 & 1 & 2 & 3 & 7 & 8 & 9 & 11 & 12 & 13 & 14 & \cdots \end{pmatrix} = g.$$

*Note that $\max A_i \notin \text{ran} f$ for all $A_i$ such that $i \geq 1$. The main ideas behind the concept are as follows: We illustrate the idea by considering $1, 3, 6,$ and $10 \notin \text{ran } f$. Since $1 \in (X_2, 1)$ and $(X_2, 1) \cap \text{ran } f = \varnothing$, $1h = 1$. Consider that $3 \in (X_2, 3)$ and $(X_2, 3) \cap \text{ran } f = \{2\}$. We can see that $1f = 2$. Hence, $y_2 = 1$ and $y_2 g = 1$.*

*Therefore, $3h = 1$. Consider that $6 \in (X_2, 6)$ and $(X_2, 6) \cap ran\, f = \{4, 5\}$. Then, we choose $4 \in (X_3, 6) \cap ran\, f$. We can see that $2f = 4$. Hence, $y_4 = 2$ and $y_4g = 2$. Therefore, $6h = 2$. Consider that $10 \in (X_2, 10)$ and $(X_2, 10) \cap ran\, f = \{7, 8, 9\}$. Then, we choose $7 \in (X_2, 10) \cap ran\, f$. We can see that $4f = 7$. Hence, $y_7 = 4$ and $y_7g = 7$. Therefore, $10h = 7$.*

**Theorem 1.** *Let $\mathcal{P} = \{X_i \mid i \in \Lambda\}$ be a partition on a set $X$ such that $X_i$ is infinite for some $i \in \Lambda$ and $X/\rho$ is a refinement of $\mathcal{P}$. A function $f$ is a left magnifier of $T_\rho(X, \mathcal{P})$ if and only if $f \in T_\rho(X, \mathcal{P})$ is injective but not surjective and for any $x, y \in X$, $(xf, yf) \in \rho$ implies $(x, y) \in \rho$.*

**Proof.** By Corollary 1, Lemmas 2 and 5. □

Alternatively, Theorem 1 can be assured by applying the result presented in [15] to the restriction of $f \in T_\rho(X)$ to the single set of $\mathcal{P}$.

**Theorem 2.** *Let $\mathcal{P} = \{X_i \mid i \in \Lambda\}$ be a partition and $\rho$ be an equivalence relation on a set $X$ and $X/\rho$ is a refinement of $\mathcal{P}$. There exists a left magnifier in $T_\rho(X, \mathcal{P})$ if and only if $X_i$ is infinite for some $i \in \Lambda$.*

**Proof.** The necessity is obtained by Lemma 4. On the other hand, suppose that there exists $X_i \in \mathcal{P}$ such that $|X_i|$ is infinite.

**Case 3.** There exists $t \in X$ such that $(X_i, t)$ is infinite. Then, there is a proper subset $A$ of $(X_i, t)$ such that $|A| = |(X_i, t)| = |(X_i, t) \setminus A|$. Thus, there is a bijection $g$ from $(X_i, t)$ to $A$. Define a function $f$ by

$$xf = \begin{cases} xg & \text{if } x \in (X_i, t), \\ x & \text{otherwise.} \end{cases}$$

Clearly, $f \in T_\rho(X, \mathcal{P})$ and $f$ is injective. Hence, $ran f \subseteq X \setminus ((X_i, t) \setminus A) \neq X$. Then, $f$ is injective but not surjective. By Theorem 1, $f$ is a left magnifier.

**Case 4.** $(X_i, t)$ is finite for all $t \in X$.

**Case 4(i):** There is a natural number $n$ such that $K = \{(X_i, t) \mid t \in X_i \text{ and } |(X_i, t)| = n\}$ is infinite. Then, there exists a proper subset $K'$ of $K$ such that $|K'| = |K| = |K \setminus K'|$. There is a bijection $\lambda$ from $K$ to $K'$. Thus, $|A| = |A\lambda| = n$ for all $A \in K$. Hence, for all $A \in K$, there exists a bijective function $\gamma_A$ from $A$ to $A\lambda$. Let $\gamma = \bigcup_{A \in K} \gamma_A$. Then, $\gamma$ is a bijection from $\bigcup_{A \in K} A$ to $\bigcup_{A \in K'} A$. Define a function $f$ by

$$xf = \begin{cases} x\gamma & \text{if } x \in \bigcup_{A \in K} A, \\ x & \text{otherwise.} \end{cases}$$

Clearly, $f$ belongs to $T_\rho(X, \mathcal{P})$ and $f$ is injective. Since $ran\, f = X \setminus (\bigcup_{A \in K} A \setminus \bigcup_{A \in K'} A) \neq X$, $f$ is not surjective. By Theorem 1, $f$ is a left magnifier.

**Case 4(ii):** For all $n \in \mathbb{N}$, the set $K = \{(X_i, t) \mid t \in X_i \text{ and } |(X_i, t)| = n\}$ is finite. Then, for each $t \in X$, there exists $t' \in X$ such that $|(X_i, t)| < |(X_i, t')|$. Let $E = \{(X_i, t) \mid [t]_\rho \subseteq X_i\}$. In this case, $E$ is an infinite set. Let $n_1 = \min_{(X_i, t) \in E} |(X_i, t)|$ and $K_1 = \{(X_i, t) \mid |(X_i, t)| = n_1\}$. Choose $(X_i, t_1) \in K_1$. Let $n_2 = \min_{(X_i, t) \in E_1} |(X_i, t)|$ where $E_1 = E \setminus K_1$ and $K_2 = \{(X_i, t) \mid |(X_i, t)| = n_2\}$. Choose $(X_i, t_2) \in K_2$. Proceeding in this way, we obtain the sets $(X_i, t_1), (X_i, t_2), \ldots, (X_i, t_k), \ldots$ and positive integers $n_1, n_2, \ldots, n_k, \ldots$ such that $n_k = \min_{(X_i, t) \in E_k} |(X_i, t)|$ where $E_k = E \setminus \bigcup_{l=1}^{k-1} K_l$ and $(X_i, t_k) \in K_k$, where $K_k = \{(X_i, t) \mid |(X_i, t)| = n_k\}$ for all $k \geq 2$. Clearly, $n_1 < n_2 < \ldots < n_k < \ldots$. Next, we let $A = \{(X_i, t_j) \mid j \geq 1\}$. Then, $|(X_i, t_j)| < |(X_i, t_{j+1})|$ for all $j \geq 1$. Hence, there exists an injection

$\gamma_j : (X_i, t_j) \to (X_i, t_{j+1})$. Let $\gamma = \bigcup_{j \geq 1} \gamma_j$. Then, $\gamma$ is an injection from $\bigcup_{B \in A} B$ to itself. Next, define a function $f$ by

$$xf = \begin{cases} x\gamma & \text{if } x \in \bigcup_{B \in A} B, \\ x & \text{otherwise.} \end{cases}$$

Clearly, $f \in T_\rho(X, \mathcal{P})$ and $f$ is injective. Since ran $f \subseteq X \setminus (X_i, t_1) \neq X$, $f$ is not surjective. By Theorem 1, $f$ is a left magnifier. $\square$

## 3. Right Magnifier in $T_\rho(X, \mathcal{P})$

In this section, we pay our attention to right magnifiers in $T_\rho(X, \mathcal{P})$. We provide the properties of right magnifiers in $T_\rho(X, \mathcal{P})$ and conditions for elements in this semigroup to be a right magnifier.

**Lemma 6.** *If $f$ is a right magnifier in $T_\rho(X, \mathcal{P})$, then $f$ is surjective.*

**Proof.** By assumption, let $M \subsetneq T_\rho(X, \mathcal{P})$ be such that $Mf = T_\rho(X, \mathcal{P})$. Clearly, the identity map $id_X$ on $X$ belongs to $T_\rho(X, \mathcal{P})$. Thus, there exists $h \in M$ such that $hf = id_X$. This implies that $f$ is surjective. $\square$

**Lemma 7.** *Let $f$ be a right magnifier in $T_\rho(X, \mathcal{P})$. For any $x, y \in X$, if $(x, y) \in \rho$, then there exists $(a, b) \in \rho$ such that $x = af$ and $y = bf$.*

**Proof.** By assumption, let $M \subsetneq T_\rho(X, \mathcal{P})$ be such that $Mf = T_\rho(X, \mathcal{P})$. Then, $hf = id_X$ for some $h \in M$. Let $x, y \in X$ be such that $(x, y) \in \rho$. Then, $xhf = xid_X = x$ and $yhf = yid_X = y$. Since $h \in T_\rho(X, \mathcal{P})$, $(xh, yh) \in \rho$. Choose $a = xh$, $b = yh$. Therefore, $(a, b) \in \rho$ such that $x = af$ and $y = bf$. $\square$

Note that, for all surjection $f \in T_\rho(X, \mathcal{P})$, each $x \in X_i$ with $X_i \in \mathcal{P}$, there exists $a \in X_i$ such that $af = x$. Consequently, any right magnifier has this property, by Lemma 6.

**Lemma 8.** *Let $f$ be an element in $T_\rho(X, \mathcal{P})$. If $f$ is bijective, then $f$ is not a right magnifier of $T_\rho(X, \mathcal{P})$.*

**Proof.** Let $f$ be a bijective function in $T_\rho(X, \mathcal{P})$. Suppose that $f$ is a right magnifier. Then, there is a proper subset $M$ of $T_\rho(X, \mathcal{P})$ such that $Mf = T_\rho(X, \mathcal{P})$. Clearly, $Mf \subseteq T_\rho(X, \mathcal{P})f$ and $T_\rho(X, \mathcal{P})f \subseteq T_\rho(X, \mathcal{P}) = Mf$. Therefore, $Mf = T_\rho(X, \mathcal{P})f$. Since $f$ is injective, $M = T_\rho(X, \mathcal{P})$, which is a contradiction. $\square$

By Lemmas 6 and 8, we obtain Corollary 3.

**Corollary 3.** *If $f$ is a right magnifier in $T_\rho(X, \mathcal{P})$, then $f$ is surjective but not injective.*

**Lemma 9.** *Let $\mathcal{P} = \{X_i \mid i \in \Lambda\}$ be a partition on a set $X$. If $X_i$ is finite for all $i \in \Lambda$, then there exists no right magnifier in $T_\rho(X, \mathcal{P})$.*

**Proof.** Suppose that there is a right magnifier $f \in T_\rho(X, \mathcal{P})$. By assumption and Lemma 7, $f|_{X_i}$ is an injection for all $i \in \Lambda$. Since $Xf = \left( \bigcup_{i \in \Lambda} X_i \right)f = \bigcup_{i \in \Lambda} X_i f$, $f$ is injective, which is a contradiction. $\square$

It is noticeable in Lemma 9 that, if a right magnifier exists in $T_\rho(X, \mathcal{P})$, then $X_i$ is infinite for some $i \in \Lambda$. However, the converse of this statement is not true in general. It is illustrated by the following counterexample.

**Example 4.** *Let $X = \mathbb{Z}$ and $\mathcal{P} = \{X_i \mid i \in \mathbb{N} \cup \{0\}\}$ where $X_0 = \{0, -1, -2, \ldots\}$ and $X_i = \{2i - 1, 2i\}$ for all $i \in \mathbb{N}$. Define a relation $\rho$ on $X$ by $\rho = \bigcup_{j=1}^{\infty} (A_j \times A_j)$ where $A_1 = \{0, \pm1, \pm2\}$ and $A_j = \{\pm 2j, \pm(2j-1)\}$ for all $j \in \mathbb{N}$ and $j \geq 2$. Clearly, $(X_i, x)$ is finite for all $i \in \mathbb{N} \cup \{0\}$ and $x \in X$. Moreover, $A_i = (X_0, x_i) \sqcup X_i$ where $[x_i]_\rho = A_i$ for all $i \in \mathbb{N}$. Note that $(X_0, x_i) \subseteq A_i$ where $[x_i] = A_i$ and $X_i \subseteq A_i$ for all $i \in \mathbb{N}$. Let $f$ be a surjection in $T_\rho(X, \mathcal{P})$. Then, $X_i f = X_i \subseteq A_i$ for all $i \in \mathbb{N}$. This forces that $(X_0, x_i) f = (X_0, x_i) \subseteq A_i$ where $[x_i]_\rho = A_i$ for all $j \in \mathbb{N}$. Then, $f$ is an injection. This implies that there exists no right magnifier in $T_\rho(X, \mathcal{P})$.*

**Corollary 4.** *If $X$ is a finite set, then $T_\rho(X, \mathcal{P})$ has no right magnifiers.*

**Lemma 10.** *Let $\mathcal{P} = \{X_i \mid i \in \Lambda\}$ be a partition on a set $X$. If $f \in T_\rho(X, \mathcal{P})$ is surjective but not injective and for any $(x, y) \in \rho$, there exists $(a, b) \in \rho$ such that $x = af$ and $y = bf$, then $f$ is a right magnifier.*

**Proof.** Assume that $f \in T_\rho(X, \mathcal{P})$ is surjective but not injective. Let $g \in T_\rho(X, \mathcal{P})$ and $M = \{h \in T_\rho(X, \mathcal{P}) \mid h \text{ is not a surjection }\}$. Let $x \in (X_i, x_j)$. Thus, $xg \in X_i$. Since $f$ is surjective, we can choose $y_x \in X_i$ such that $y_x f = xg$. Then, for all $x, z \in (X_i, x_j)$, there are $y_x, y_z \in X_i$ such that $y_x f = xg$, $y_z f = zg$ and $(y_x, y_z) \in \rho$ because $(x, z) \in \rho$. Define a function $h$ by $xh = y_x$ for all $x \in X$. Obviously, $h$ is a function on $X$. To show that $h \in T_\rho(X, P)$, let $a, b \in X$ be such that $(a, b) \in \rho$. Clearly, $(ah, bh) = (y_a, y_b) \in \rho$. For each $X_i \in \mathcal{P}$, if $a \in X_i$, then $ah = y_a$. Since $y_a f = a$ and $ah \in X_i$, $h \in T_\rho(X, \mathcal{P})$. Since $f$ is not injective, there are distinct two elements $x, y \in X_i$ for some $i \in \Lambda$ such that $xf = yf = z$ for some $z \in X_i$. Hence, $y_z = x$ or $y_z = y$. Therefore, either $x \notin \operatorname{ran} h$ or $y \notin \operatorname{ran} h$. Thus, $h \in M$. For all $x \in X$, $xhf = y_x f = xg$. Then, $hf = g$. Therefore, $Mf = T_\rho(X, \mathcal{P})$ which means that $f$ is a right magnifier.  □

The ideas of Lemma 10 are illustrated by the next example.

**Example 5.** *Let $X = \mathbb{Z}$ and $\mathcal{P}$ be a partition on $X$ such that $\mathcal{P} = \{\mathbb{Z}^-, \mathbb{Z}^+ \cup \{0\}\}$. Define a relation $\rho$ on $X$ by $\rho = \bigcup_{j=1}^{\infty} (A_j \times A_j)$ where $A_1 = \{-1, -3, -5, -7, \ldots, \}$, $A_2 = \{\pm2, \pm4, \pm6, \pm8, \ldots\}$, $A_3 = \{0\}$, $A_4 = \{1, 3\}$, $A_5 = \{5, 7\}$, ... Then, $\rho$ is an equivalence relation on $X$ and $X/\rho = \{\{-1, -3, -5, -7, \ldots, \}, \{\pm2, \pm4, \pm6, \pm8, \ldots\}, \{0\}, \{1, 3\}, \{5, 7\}, \ldots\}$. Let $f$ be a function defined by $xf = 4$ for all $x \in A_4$ and*

$$xf = \begin{cases} x - 4 & \text{if } x \in \mathbb{Z}^+ \setminus 2\mathbb{Z}^+, \\ x + 4 & \text{if } x \in 4\mathbb{Z}^+, \\ x & \text{otherwise.} \end{cases}$$

*It is easy to see that $f$ belongs to $T_\rho(X, \mathcal{P})$, and it is injective but not surjective. However, we can not construct the function $h \in T_\rho(X, \mathcal{P})$ such that $fh = id_X$ since $(4, 8) \in \rho$ but there exist no $a, b \in X$ such that $(a, b) \in \rho$ satisfies $af = 4$ and $bf = 8$.*

**Example 6.** *Let $X = \mathbb{N}$ and $\mathcal{P}$ be a partition on $X$ such that $\mathcal{P} = \{\{1\}, \{2\}, \{3, 4, 5\}, \{6, 7, 8, 9, \ldots\}\}$. Define a relation $\rho$ on $X$ by*

$$x\rho y \text{ if and only if } x \equiv y \mod 2.$$

*It is obvious that $\rho$ is an equivalence relation on $X$. We obtain that $X/\rho = \{\{1, 3, 5, 7, \ldots\}, \{2, 4, 6, 8, \ldots\}\}$ and $\{6, 7, 8, 9, \ldots\} \in \mathcal{P}$ is infinite. Let $f$ be a function defined by $1f = 1, 2f = 2, 3f = 5, 4f = 4, 5f = 3, 6f = 6, 7f = 7$ and $xf = x - 2$ for all $x \geq 8$. Thus, we write $f$ as*

$$f = \begin{pmatrix} 1 & 2 & 3 & 4 & 5 & 6 & 7 & 8 & 9 & 10 & 11 & 12 & 13 & 14 & 15 & \cdots \\ 1 & 2 & 5 & 4 & 3 & 6 & 7 & 6 & 7 & 8 & 9 & 10 & 11 & 12 & 13 & \cdots \end{pmatrix}.$$

*It is easy to see that $f$ belongs to $T_\rho(X, \mathcal{P})$ and it is surjective but not injective. Let $M = \{h \in T_\rho(X, \mathcal{P}) \mid h$ is not a surjection$\}$ and consider the element $g$ of $T_\rho(X, \mathcal{P})$, which is defined by $1g = 1, 2g = 2, 3g = 5, 4g = 4, 5g = 3$, and $xf = x + 2$ for all $x \geq 6$.*

*Thus, we write $g$ as*

$$g = \begin{pmatrix} 1 & 2 & 3 & 4 & 5 & 6 & 7 & 8 & 9 & 10 & 11 & 12 & 13 & 14 & 15 & \cdots \\ 1 & 2 & 5 & 4 & 3 & 8 & 9 & 10 & 11 & 12 & 13 & 14 & 15 & 16 & 13 & \cdots \end{pmatrix}.$$

*By Lemma 10, $f$ is a right magnifier and then there exists a function $h \in M$ such that $hf = g$. To get the desired result, define a function $h$ in $T_\rho(X, \mathcal{P})$ by*

$$xh = \begin{cases} x & \text{if } x \leq 5, \\ x + 4 & \text{if } x > 5. \end{cases}$$

*Thus, we write $h$ as*

$$h = \begin{pmatrix} 1 & 2 & 3 & 4 & 5 & 6 & 7 & 8 & 9 & 10 & 11 & 12 & 13 & 14 & 15 & \cdots \\ 1 & 2 & 3 & 4 & 5 & 10 & 11 & 12 & 13 & 14 & 15 & 16 & 17 & 18 & 19 & \cdots \end{pmatrix}.$$

*Clearly, $h \in M$. Therefore,*

$$hf = \begin{pmatrix} 1 & 2 & 3 & 4 & 5 & 6 & 7 & 8 & 9 & 10 & 11 & 12 & 13 & 14 & 15 & \cdots \\ 1 & 2 & 3 & 4 & 5 & 10 & 11 & 12 & 13 & 14 & 15 & 16 & 17 & 18 & 19 & \cdots \end{pmatrix}$$

$$\begin{pmatrix} 1 & 2 & 3 & 4 & 5 & 6 & 7 & 8 & 9 & 10 & 11 & 12 & 13 & 14 & 15 & \cdots \\ 1 & 2 & 5 & 4 & 3 & 6 & 7 & 6 & 7 & 8 & 9 & 10 & 11 & 12 & 13 & \cdots \end{pmatrix}$$

$$= \begin{pmatrix} 1 & 2 & 3 & 4 & 5 & 6 & 7 & 8 & 9 & 10 & 11 & 12 & 13 & 14 & 15 & \cdots \\ 1 & 2 & 5 & 4 & 3 & 8 & 9 & 10 & 11 & 12 & 13 & 14 & 15 & 16 & 13 & \cdots \end{pmatrix} = g.$$

**Theorem 3.** *Let $\mathcal{P} = \{X_i \mid i \in \Lambda\}$ be a partition on a set $X$. A function $f$ is a right magnifier of $T_\rho(X, \mathcal{P})$ if and only if $f$ is surjective but not injective and for any $(x, y) \in \rho$, there exists $(a, b) \in \rho$ such that $x = af$ and $y = bf$.*

**Proof.** By Corollary 3, Lemmas 7 and 10. □

**Theorem 4.** *Let $\mathcal{P}$ be a partition and $\rho$ be an equivalence relation on a set $X$ such that $X/\rho$ is a refinement of $\mathcal{P}$. There exists a right magnifier in $T_\rho(X, \mathcal{P})$ if and only if $X_i$ is infinite for some $i \in \Lambda$.*

**Proof.** The necessity is obtained by Lemma 9. On the other hand, suppose that there exists $X_i \in \mathcal{P}$ such that $|X_i|$ is infinite.

**Case 5.** There exists $t \in X$ such that $(X_i, t)$ is infinite. Then, there is a proper subset $A$ of $(X_i, t)$ such that $|A| = |(X_i, t)| = |(X_i, t) \setminus A|$. Thus, there is a bijective function $g$ from $A$ to $(X_i, t)$. Define a function $f$ by

$$xf = \begin{cases} xg & \text{if } x \in A, \\ x & \text{otherwise.} \end{cases}$$

Clearly, $f \in T_\rho(X, \mathcal{P})$ and $f$ is surjective. Since $(X \setminus (X_i \setminus A))f = X$, $f$ is not injective. By Theorem 3, $f$ is a right magnifier.

**Case 6.** $(X_i, t)$ is finite for all $t \in X$.

**Case 6(i).** There is a natural number $n$ such that $K = \{(X_i, t) \mid t \in X_i$ and $|(X_i, t)| = n\}$ is infinite. Then, there exists a proper subset $K'$ of $K$ such that $|K'| = |K| = |K \setminus K'|$. There is a bijective function

$\lambda$ from $K'$ to $K$. Thus, $|A| = |A\lambda| = n$ for all $A \in K'$. Hence, for all $A \in K'$, there exists a bijective function $\gamma_A$ from $A$ to $A\lambda$. Let $\gamma = \bigcup_{A \in K'} \gamma_A$. Then, $\gamma$ is a bijection from $\bigcup_{A \in K'} A$ to $\bigcup_{A \in K} A$. Define a function $f$ by

$$xf = \begin{cases} x\gamma & \text{if } x \in \bigcup_{A \in K'} A, \\ x & \text{otherwise.} \end{cases}$$

Clearly, $f$ belongs to $T_\rho(X, \mathcal{P})$ and $f$ is surjective. Since $(X \setminus (\bigcup_{A \in K} A \setminus \bigcup_{A \in K'} A))f = X$, $f$ is not injective. By Theorem 3, $f$ is a right magnifier.

**Case 6(ii).** For all $n \in \mathbb{N}$, the set $K = \{(X_i, t) \mid t \in X_i \text{ and } |(X_i, t)| = n\}$ is finite. Then, for each $t \in X$, there exists $t' \in X$ such that $|(X_i, t)| < |(X_i, t')|$. Let $E = \{(X_i, t) \mid [t]_\rho \subseteq X_i\}$. In this case, $E$ is an infinite set. Let $n_1 = \min_{(X_i, t) \in E} |(X_i, t)|$ and $K_1 = \{(X_i, t) \mid |(X_i, t)| = n_1\}$. Choose $(X_i, t_1) \in K_1$. Let $n_2 = \min_{(X_i, t) \in E_1} |(X_i, t)|$ where $E_1 = E \setminus K_1$ and $K_2 = \{(X_i, t) \mid |(X_i, t)| = n_2\}$. Choose $(X_i, t_2) \in K_2$. Proceeding in this way, we obtain the sets $(X_i, t_1), (X_i, t_2), \ldots, (X_i, t_k), \ldots$ and positive integers $n_1, n_2, \ldots, n_k, \ldots$ such that $n_k = \min_{(X_i, t) \in E_k} |(X_i, t)|$ where $E_k = E \setminus \bigcup_{l=1}^{k-1} K_l$ and $(X_i, t_k) \in K_k$, where $K_k = \{(X_i, t) \mid |(X_i, t)| = n_k\}$ for all $k \geq 2$. Clearly, $n_1 < n_2 < \ldots < n_k < \ldots$. Next, we let $A = \{(X_i, t_j) \mid j \geq 1\}$. Then, $|(X_i, t_j)| < |(X_i, t_{j+1})|$ for all $j \geq 1$. Hence, there exists a surjection $\gamma_j : (X_i, t_j) \to (X_i, t_{j-1})$ for all $j \geq 2$. Let $\gamma = \bigcup_{j \geq 2} \gamma_j$. Then, $\gamma$ is a surjection from $\bigcup_{B \in A} B \setminus (X_i, t_1)$ to $\bigcup_{B \in A} B$. Next, define a function $f$ by

$$xf = \begin{cases} x\gamma & \text{if } x \in \bigcup_{B \in A} B \setminus (X_i, t_1), \\ x & \text{otherwise.} \end{cases}$$

Clearly, $f \in T_\rho(X, \mathcal{P})$ and $f$ is a surjection. Since $(X_i, t_1)f = (X_i, t_1) = (X_i, t_2)f$, $f$ is not injective. By Theorem 3, $f$ is a right magnifier. □

## 4. Conclusions

Let $\mathcal{P} = \{X_i \mid i \in \Lambda\}$ be a partition on a nonempty set $X$. If $X_i$ is finite for all $i \in \Lambda$, then neither a left magnifier nor a right magnifier exists in $T_\rho(X, \mathcal{P})$. Assume that $X_i$ is infinite for some $i \in \Lambda$. Each of the following statements holds true:

1. If $X/\rho$ is a refinement of $\mathcal{P}$, then $f$ is a left magnifier if and only if $f \in T_\rho(X, \mathcal{P})$ is injective but not surjective and for any $x, y \in X$, $(xf, yf) \in \rho$ implies $(x, y) \in \rho$.
2. A function $f \in T_\rho(X, \mathcal{P})$ is a right magnifier if and only if $f$ is surjective but not injective and for any $x, y \in X$, $(x, y) \in \rho$ implies $x = af$ and $y = bf$ for some $(a, b) \in \rho$.
3. Magnifiers exist in $T_\rho(X, \mathcal{P})$ if and only if $X_i$ is infinite for some $i \in \Lambda$, provided that $X/\rho$ is a refinement of $\mathcal{P}$.

**Author Contributions:** Conceptualization, T.K., M.P., and R.C.; Investigation, T.K., M.P., and R.C.; Writing—Original Draft Preparation, T.K., M. P. and R.C.; Writing—Review and Editing, T.K., M.P., and R.C.; Supervision, R.C. All authors have read and agreed to the published version of the manuscript.

**Funding:** This research was supported by the Algebra and Applications Research Unit, Department of Mathematics and Statistics, Faculty of Science, Prince of Songkla University.

**Acknowledgments:** The authors would like to express our appreciation to the anonymous referees for the comprehensive reading of this paper and their valuable comments and suggestions.

**Conflicts of Interest:** The authors declare no conflict of interest.

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
