# Peer review of "Magnifiers in Some Generalization of the Full Transformation Semigroups"

_mathematics, doi:10.3390/math8040473_

Round 1

Reviewer 1 Report

View attached document

Author Response

We provide specific answers below comments, please see in the attached file. 

The authors would like to express our appreciation for the comprehensive reading of this paper and their valuable comments and suggestions.

Reviewer 2 Report

See attached pdf (report.pdf)

Author Response

(The authors gave the same response as above.)

Reviewer 3 Report

The article deals with an interesting subject. The introduction is synthetic but informative, presenting the state of the art correctly.
The results and their demonstrations are quite clear, leaving no doubt.
The subject covered in this article is somewhat delimited, but the article is developed by studying exactly what is proposed in the initial summary.
Perhaps it would be useful to present the consequences of what has been demonstrated in anticipating future work.

Author Response

We check carefully of this paper and revise followed by reviewers's comments and suggestions. This file is the latest form of this manuscript.

Round 2

Reviewer 1 Report

The revised version of the paper follows all the referee's suggestions. Now, it deserves publication.